# RAVL: REACH-AWARE VALUE LEARNING FOR THE EDGE-OF-REACH PROBLEM IN OFFLINE MODEL-BASED REINFORCEMENT LEARNING

## ABSTRACT

Offline reinforcement learning makes use of pre-collected datasets and has emerged as a powerful paradigm for training agents without the need for expensive or unsafe online data collection. This offline approach, however, introduces the additional challenge of evaluating values for state-actions not seen in the dataset – termed the *out-of-sample* problem. Model-based approaches deal with this by allowing the agent to collect additional data through rollouts in a learned dynamics model. The prevailing theoretical understanding is that this effectively resolves the out-of-sample issue, and that any remaining difficulties are due to errors in the learned dynamics model. Based on this understanding, one would expect improvements to the dynamics model to lead to improvements to the learned policy. Surprisingly, however, we find that existing algorithms *completely fail when the true dynamics are provided in place of the learned dynamics model*. This observation exposes a common misconception in offline reinforcement learning, namely that dynamics model errors do not explain the behavior of model-based methods. Our subsequent investigation reveals a second major and previously overlooked issue in offline model-based reinforcement learning (which we term the *edge-of-reach* problem), whereby values of states that are only reachable in the final step of the limited horizon rollouts are pathologically overestimated, similar to the *out-of-sample* problem faced by model-free methods. This new insight fills some of the gaps in existing theory and allows us to reinterpret the efficacy of prior model-based methods. Guided by this understanding, we propose *Reach-Aware Value Learning* (RAVL), a value-based algorithm that is able to capture value uncertainty at edge-of-reach states. Our method achieves strong performance on the standard D4RL benchmark, and we hope that the insights developed in this paper aid the future design of more accurately motivated offline algorithms.

## 1 INTRODUCTION

Offline reinforcement learning (Offline RL, Ernst et al. (2005); Levine et al. (2020)) allows agents to learn from pre-existing datasets. This makes it a practical choice for scenarios where online data collection could be costly or dangerous, such as robotics (Ball et al., 2021; Chebotar et al., 2021; Kumar et al., 2021) or healthcare (Shiranthika et al., 2022; Tang & Wiens, 2021). Furthermore, offline RL provides a route for utilizing large existing datasets to scale RL to more complex problems (Kumar et al., 2023), mirroring recent advances in supervised learning (Brown et al., 2020).

The central challenge in offline RL is dealing with value estimation for counterfactual actions not present in the dataset (termed the *out-of-sample* problem (Kostrikov et al., 2022)). A naïve approach results in inaccurate out-of-distribution values being propagated through to observed state-actions, which can lead to pathological training dynamics (Kumar et al., 2019). There have been many proposals to resolve this, with approaches largely falling into one of two categories: model-free (An et al., 2021; Fujimoto & Gu, 2021; Fujimoto et al., 2019; Kostrikov et al., 2022; Kumar et al., 2019; 2020) or model-based (Kidambi et al., 2020; Lu et al., 2022; Sun et al., 2023; Yu et al., 2020).

Model-free methods typically address the out-of-sample problem by applying a form of conservatism or constraint to avoid out-of-sample state-actions in the Bellman update. In contrast, model-

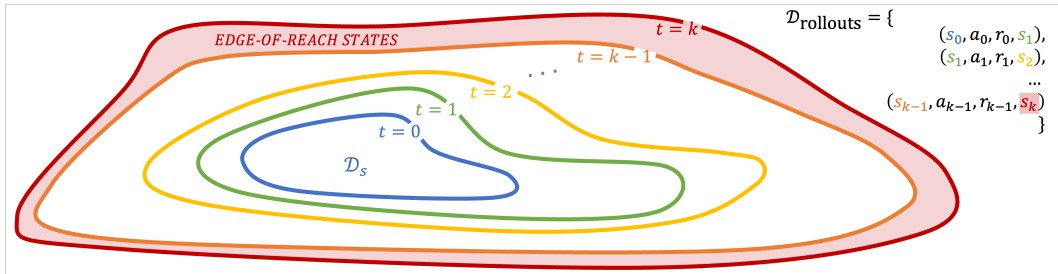

Figure 1: Our paper investigates the overlooked *edge-of-reach* problem in offline model-based reinforcement learning, whereby states (in **red**) only reachable at the final step of a model rollout are liable to pathological value overestimation. We show the boundaries of reachable states at each timestep of $k$-step rollouts from $\mathcal{D}_s$.

based methods (Kidambi et al., 2020; Sun et al., 2023; Yu et al., 2020), which are the focus of this paper, aim to solve the out-of-sample problem by using a learned dynamics model (Janner et al., 2019; Sutton, 1991) to allow the agent to collect additional observations for previously out-of-sample state-actions. The current understanding is that these additional rollouts allow the agent to collect corrective feedback for any misestimated values, thereby solving the out-of-sample problem analogously to how it is avoided in online RL. With this understanding, model-based methods typically attribute any remaining difficulties to errors in the learned dynamics model, and algorithms are subsequently motivated by avoiding model exploitation using some notion of dynamics uncertainty.

This understanding of model-based approaches naturally leads to the belief that improving the dynamics model should lead to stronger performance. However, in this paper, we begin by demonstrating the surprising observation that existing offline model-based methods completely fail when trained with the true error-free dynamics model (see Table 1). This exposes *a gap in the current predominant understanding of model-based offline RL methods*. From our subsequent investigation, we find that the combination of a *fixed dataset with limited-horizon rollouts* results in there being a set of states which, under any policy, are only reachable in the final rollout step. For these *edge-of-reach* states, even with the ability to collect additional data, the agent is never able to observe the outcome of any actions from them. The result is pathological value overestimation closely related to that seen with the out-of-sample problem in model-free methods. We term this the *edge-of-reach* problem. Thus, contrary to common understanding, the out-of-sample problem key in model-free methods can be seen to effectively persist in model-based methods.

We illustrate this problem with careful and thorough analysis on a simple environment. We demonstrate first, that pathological value overestimation can occur despite allowing additional data collection, and second, that correcting *only the value estimates of edge-of-reach states* is sufficient to resolve this. This leads us to propose *Reach-Aware Value Learning* (RAVL), a scalable solution based on value ensembles that are able to capture the value uncertainty at edge-of-reach states. Our method connects the two previously disjoint subfields of model-free and model-based offline RL algorithms, combining the strengths of both. We show that RAVL provides a satisfying solution on the simple environment, and moreover scales to the standard D4RL (Fu et al., 2020) benchmark.

The insights presented in this paper allow us to fill some of the gaps in our existing understanding, and subsequently enable us to reinterpret the success of prior offline model-based RL algorithms based on this more complete understanding of the underlying mechanisms. Furthermore, our findings serve to unify understanding of model-based and model-free algorithms, and as such may pave the way for porting further strengths of existing model-free algorithms into model-based approaches. To summarize, our contributions are as follows:

- We expose a gap in the current understanding of offline model-based RL algorithms using experiments with the oracle dynamics model in Section 4.1. This leads us to identify the existence of *edge-of-reach* states.

- We describe the edge-of-reach problem in Section 4.2 and carefully demonstrate the phenomenon in a simple environment in Section 5, including experiments to verify that failure stems from value overestimation of edge-of-reach states.

- In light of this new understanding, we propose *Reach-Aware Value Learning* (RAVL), a natural solution to the edge-of-reach problem in Section 6. We show that our algorithm provides an effective solution on the simple environment and also scales to achieve strong performance on the standard offline RL benchmark, D4RL (Fu et al., 2020).

## 2 BACKGROUND

### 2.1 REINFORCEMENT LEARNING AND VALUE FUNCTIONS

We consider the standard RL framework (Sutton & Barto, 2018), in which the environment is formulated as a Markov Decision Process, $M = (\mathcal{S}, \mathcal{A}, T, R, \mu_0, \gamma)$, where $\mathcal{S}$ and $\mathcal{A}$ denote the state and action spaces, $T(s'|s, a)$ and $R(s, a)$ denote the transition and reward dynamics, $\mu_0$ the initial state distribution, and $\gamma \in (0, 1)$ is the discount factor. The goal in reinforcement learning is to learn a policy $\pi(a|s)$ that maximizes the expected discounted return:

$$\pi^* = \arg\max_{\pi} \mathbb{E}\left[\sum_{t=0}^{\infty} \gamma^t R(s_t, a_t) \mid s_0 \sim \mu_0(\cdot), a_t \sim \pi(\cdot \mid s_t), s_{t+1} \sim T(\cdot \mid s_t, a_t)\right]. \quad (1)$$

The broad class of algorithms we consider are actor-critic (Konda & Tsitsiklis, 1999) methods which jointly learn a policy $\pi$ and state-action value function ($Q$-function). The $Q$-function $Q(s, a)$ aims to predict the expected discounted return conditional on starting at state $s$, taking action $a$, and henceforth following the current policy. We then iterate between improving the policy relative to the current $Q$-function (policy improvement) and fitting the $Q$-function to the current policy (policy evaluation) to learn the optimal policy. The policy evaluation step relies on approximate dynamic programming with Bellman updates, updating the $Q$-function using some dataset $\mathcal{D}$ as follows:

$$Q^{j+1} \leftarrow \arg\min_{Q} \mathbb{E}_{(s,a,r,s')\sim\mathcal{D}, a'\sim\pi^j(\cdot|s')}[(\underbrace{Q(s, a)}_{\text{input}} - \underbrace{[r + \gamma Q^j(s', a')]}_{\text{Bellman target}})^2] \quad (2)$$

### 2.2 OFFLINE REINFORCEMENT LEARNING AND THE OUT-OF-SAMPLE PROBLEM

In *online* RL, the agent is able to collect new on-policy data throughout training. By contrast, in the *offline* setting, the algorithm has access only to a fixed offline dataset consisting of transition tuples $\mathcal{D}_{\text{offline}} = \{(s_i, a_i, r_i, s'_i)\}_{i=1,\ldots,N}$ collected by one or more behavioral policies $\pi^\beta$. The central problem in offline learning is, therefore, the counterfactual action or *out-of-sample* problem:[1] where state-actions pairs $(s', a')$ used to compute the targets in the Bellman update (see Equation (2)) may not appear in the dataset $\mathcal{D}_{\text{offline}}$. The consequence is that these *out-of-sample* state-actions are never optimized due to never appearing as inputs in the Bellman update, meaning they are prone to being erroneously misestimated. Further, when coupled with the max operator in policy improvement, this misestimation means naïve application of online algorithms often results in catastrophic training dynamics and pathological overestimation of $Q$-values (Kumar et al., 2019).

### 2.3 OFFLINE MODEL-BASED METHODS

Model-based methods (Sutton, 1991) aim to solve the out-of-sample issue by allowing the agent to collect additional synthetic data in a learned dynamics model $\widehat{M} = (\mathcal{S}, \mathcal{A}, \widehat{T}, \widehat{R}, \mu_0, \gamma)$. $\widehat{T}(s'|s, a)$ and $\widehat{R}(s, a)$ denote the approximate transition and reward functions, and are commonly realized as a deep ensemble (Chua et al., 2018; Lakshminarayanan et al., 2017). Methods typically use the MBPO procedure (Janner et al., 2019) (see Algorithm 1), with synthetic data sampled as $k$-step trajectories (termed rollouts) under the current policy, starting from states in the offline dataset $\mathcal{D}_{\text{offline}}$.

## 3 RELATED WORK

**Overestimation bias in $Q$-learning.** $Q$-learning is a popular basis for many off-policy RL algorithms, however, it has been shown to suffer from an overestimation bias stemming from the maximum operation $\max_{a' \in \mathcal{A}} Q(s', a')$ in the Bellman update (van Hasselt et al., 2016). This problem is exacerbated when combined with function approximation and bootstrapping, commonly known as the deadly triad (van Hasselt et al., 2018) and can become even more problematic in the offline

---

[1]This is often referred to as the problem of evaluating values for 'out-of-distribution' state-actions (due to a distribution shift between the policy used to sample the action and the behavior policy used to collect $\mathcal{D}_{\text{offline}}$). We choose to use the term 'out-of-sample' to highlight connections later on, but assuming that function approximators generalize within-distribution, these are essentially interchangeable.

setting due to the out-of-sample problem (Kumar et al., 2019). An effective approach to mitigating this issue is clipped double-Q learning (Fujimoto et al., 2018) (extending the original double-Q learning (van Hasselt et al., 2016)) which proposes to learn two independent $Q$-functions and take the minimum over them. Furthermore, this approach can be extended with a larger ensemble of Q-functions to further control overestimation bias (An et al., 2021; Chen et al., 2021). As discussed to in Section 2.2, overestimation bias and the subsequent pathological training dynamics is the main barrier to transferring online RL algorithms to the offline setting, due to the lack of corrective feedback on out-of-sample state-actions. As such, there have been many attempts to address it.

**Offline model-free methods.** Offline model-free algorithms can broadly be divided by how they deal with out-of-sample state-action pairs into action sampling-based methods and value pessimism-based methods. Action sampling-based methods ensure that any state-actions used in the Bellman update are close to the behavior policy $\pi^\beta$, such that they are in-distribution with respect to $\mathcal{D}_{\text{offline}}$. This may be done by updating in a SARSA (Sutton & Barto, 2018) fashion (Kostrikov et al., 2022), by modeling $\pi^\beta$ directly (Fujimoto et al., 2019), or by explicitly constraining the policy using some measure of distance to $\pi_\beta$ (Kumar et al., 2019). A related idea is to add a behavioral cloning loss term (Fujimoto & Gu, 2021). On the other hand, value pessimism-based methods aim to regularize the $Q$-function directly such that it produces low-value estimates for out-of-sample state-actions (An et al., 2021; Kostrikov et al., 2021; Kumar et al., 2020). Of these, the most similar to our RAVL algorithm is EDAC (An et al., 2021) which encourages values at out-of-distribution state-actions to be low by minimizing over a $Q$-ensemble.

**Offline model-based methods.** As alluded to in Section 1, the prevailing view in the offline model-based literature is that the ability to collect additional data in the model effectively resolves the out-of-sample issue, as the agent is now able to observe the outcome of previously out-of-sample state-actions. Thus, the primary concern stated for model-based methods is to account for errors in the learned dynamics model in order to avoid model exploitation. A broad class of methods explicitly penalizes reward by dynamics uncertainty (Kidambi et al., 2020; Lu et al., 2022; Sun et al., 2023; Yu et al., 2020), typically using variance over a dynamics ensemble. On the other hand, Rigter et al. (2022) defines a two-player game between the policy and model, updating the model in order to minimize the learned value function. This forces the policy to act conservatively in areas not covered by the dataset, where model errors may be high. Matsushima et al. (2021) borrow ideas from model-free approaches and use policy constraints, with the motivation that the model is likely to be more accurate close to the behavior policy. Critically, all previously described methods draw their motivation from dealing with dynamics model errors. Finally, most related to our method, COMBO (Yu et al., 2021), penalizes value estimates for state-actions outside model rollouts. However, similarly to Yu et al. (2020), COMBO is theoretically motivated by the assumption that one can perform infinite horizon model rollouts, which we later show overlooks serious implications.

## 4  THE EDGE-OF-REACH PROBLEM

In the following section, we describe the underlying motivation and focus of our paper: the *edge-of-reach* problem. We begin by presenting a surprising result using the ground truth dynamics, from which it follows. Next, we provide intuition for our hypothesis and theoretical proof of its effect on offline model-based training. Finally, we connect the problem to the standard *out-of-sample* issue in Appendix A, resulting in a unified view of model-free and model-based offline RL.

### 4.1  A SURPRISING RESULT: FAILURE WITH THE TRUE DYNAMICS MODEL

On the surface, the ability to collect additional data in model-based offline methods resolves the out-of-sample issue in offline RL (see Section 2.2). The key remaining difference compared to online RL therefore appears to be that: in online RL, data is collected with the true dynamics, while in offline model-based RL, data is collected in an approximate dynamics model. As a result, most existing offline model-based algorithms aim to mitigate the effect of dynamics model errors, for example by penalizing using estimates of dynamics uncertainty (Sun et al., 2023; Yu et al., 2020) or terminating after transitioning past a certain level of uncertainty (Kidambi et al., 2020).

Such an understanding would naturally lead us to expect that the ideal case would be to have a perfect model with zero dynamics error. Surprisingly, we show this is not the case. In Table 1, we

Table 1: We show, surprisingly, that if the true error-free dynamics are used in place of the learned dynamics model, existing offline model-based algorithms fail (**highlighted**) on the D4RL MuJoCo v2 datasets. Our baseline is MOPO (tuned results taken from Sun et al. (2023)). **Oracle** refers to replacing the learned dynamics model in MOPO with the true dynamics function, making it equivalent to MBPO (the base optimizer for most other offline model-based RL methods) with a perfect uncertainty-free model. We summarize this in Algorithm 1. Normalized mean and standard deviation are shown over 6 random seeds.

| | **Hopper** | | **Walker2d** | | **HalfCheetah** | |
|---|---|---|---|---|---|---|
| | **MOPO** | **Oracle** | **MOPO** | **Oracle** | **MOPO** | **Oracle** |
| random | 31.7 | 9.5±1.1 | 7.4 | -0.4±0.2 | 38.5 | 35.2±3.4 |
| medium | 62.8 | 17.2±11.1 | 84.1 | 7.2±1.6 | 73.0 | 72.9±8.6 |
| mixed | 103.5 | 71.5±32.2 | 85.6 | 7.9±1.9 | 72.1 | 72.2±4.1 |
| medexp | 81.6 | 6.1±6.8 | 112.9 | 7.7±2.1 | 90.8 | 84.9±17.6 |

demonstrate this on the D4RL (Fu et al., 2020) datasets. When the learned dynamics model used in MOPO (Yu et al., 2020) is replaced with the true dynamics (labeled **Oracle**), *most environments fail to train at all*. Furthermore, we note that other dynamics penalty-based offline model-based RL algorithms (Kidambi et al., 2020; Sun et al., 2023) share the same base MBPO (Janner et al., 2019) optimizer, hence this result indicates **the failure of all existing methods**. We provide a formal comparison between the methods used here and elsewhere in the paper in Appendix F.2.

## 4.2 THE EDGE-OF-REACH HYPOTHESIS

Our observation in Section 4.1 exposes a widespread misconception as to the true cause of issues in offline model-based RL. In the following section, we informally describe our resulting hypothesis. In offline model-based RL, synthetic data is collected by generating $k$-step rollouts. Crucially, $k$ (typically $\leq 5$) is less than the true horizon of the environment ($H = 1000$ in MuJoCo (Todorov et al., 2012)), and all rollouts begin from states sampled from the fixed offline dataset $\mathcal{D}_{\text{offline}}$. Since $\mathcal{D}_{\text{offline}}$ is typically limited, $k$-steps away from the dataset is unlikely to sufficiently cover the full state space $\mathcal{S}$. Consider states on the boundary of the space that rollouts are able to cover, i.e. states which, under any policy, are only reachable in the final step of a rollout. For these states, even with the ability to collect additional data, the agent is never able to observe the outcome of actions from them. We term these *edge-of-reach* states (see **red** in Figure 1).

More concretely, denoting the rollout-augmented dataset $\mathcal{D}_{\text{rollouts}} = \{(s_i, a_i, r_i, s_i', a_i')\}_{i=1,...,M}$, edge-of-reach states are those that appear as $s_i'$ but never, under any policy, appear as $s_i$. Crucially, this means they are used to compute the Bellman targets in $Q$-learning, but never appear as inputs to the $Q$-function and are hence themselves never updated. This makes them liable to erroneous misestimation, which bootstrapping then propagates to previous state-actions, leading to poor value estimates more widely. More significantly still, this misestimation coupled with the overestimation bias in $Q$-learning can lead to pathological overestimation, with $Q$-values being driven upwards to arbitrarily high values over training. We demonstrate this in Section 5 and Appendix B.2, and term this the *edge-of-reach* problem. In short:

> **Edge-of-Reach Hypothesis**: Limited horizon rollouts from a fixed offline dataset lead to the existence of edge-of-reach states which, due to lack of corrective feedback, can cause pathological value overestimation.

This is closely related to the out-of-sample problem in the model-free case, but focusing on the state, rather than the action part of the state-action pair. Thus, contrary to common understanding, the out-of-sample problem, which is key in model-free methods, can be seen to effectively persist in model-based methods. We provide a more thorough comparison of the out-of-sample and edge-of-reach problems in Appendix A, separating the problems into conditions required on states and actions independently. This serves to provide a unified view of model-free and model-based approaches.

## 4.3 FORMALIZATION

We now formalize the intuitions presented in Section 4.2 and present theoretical grounding for the edge-of-reach problem. Furthermore, we prove how errors can propagate to all transitions.

**Definition 1** (Edge-of-reach states). *Consider a deterministic transition model $T : \mathcal{S} \times \mathcal{A} \to \mathcal{S}$, a policy $\pi : \mathcal{S} \to \mathcal{A}$, a rollout length $k$, and a distribution over starting states $v_0(s)$, which in our case is the state distribution of $\mathcal{D}_{offline}$. Given a starting state $s_0 \sim v_0$, $k$-step rollouts are generated according to $a_t \sim \pi(\cdot|s_t)$ and $s_{t+1} \sim T(\cdot|s_t, a_t)$ for $t = 0 \ldots k-1$, giving $(s_0, a_0, s_1, \ldots, s_k)$. Let us use $p_{t,\pi}(s)$ to denote the marginal distributions over $s_t$.*

*We define a state $s \in S$ **edge-of-reach** with respect to $T$, $v_0$, and $k$ if: for $t = k$, $\exists \pi$ s.t. $p_{t,\pi}(s) > 0$, but, for $t = 1, \ldots, k-1$ and $\forall \pi$, $p_{t,\pi}(s) = 0$.*

With stochastic transition models (e.g. Gaussian models), we may have the case that no state will truly have zero density, and we relax this definition slightly to $p_{t,\pi}(s) < \epsilon$ for some small $\epsilon$. The next proposition quantifies how errors in edge-of-reach states are propagated to preceding states.

**Proposition 1.** *[Error propagation from edge-of-reach states] Consider a rollout of length $k$, $(s_0, a_0, s_1, \ldots, s_k)$. Suppose that the state $s_k$ is edge-of-reach and the approximate value function $Q^{j-1}(s_k, \pi(s_k))$ has error $\epsilon$. Then, standard value iteration will compound error $\gamma^{k-t}\epsilon$ to the estimates of $Q^j(s_t, a_t)$ for $t = 1, \ldots, k-1$. Proof provided in Appendix E.*

This is an analogous statement to that of Kumar et al. (2019); however, we highlight its significance in the context of model-based methods. While we show this for a single rollout, in practice, this behavior will manifest across sampled minibatches. We argue that this issue is the predominant issue in offline model-based algorithms, since existing works (Janner et al., 2019; Lu et al., 2022) have shown that modern dynamics models are typically accurate even to high rollout lengths.

## 5 Analysis with a Simple Environment

In the previous section, we provided a plausible theoretical explanation and intuition for the pathology found in Section 4.1. In this section, we empirically confirm this hypothesis with careful analysis on a simple environment. First, we show that naïve application of actor-critic algorithms results in exploding $Q$-values and failure to learn despite using the true dynamics. Next, we verify that edge-of-reach states are the source of this problem by correcting value estimates only at these states.

### 5.1 Setup

We consider a simple 2D continuous grid world with a reward given according to an agent's distance from a fixed goal (shown in Figure 2(a)). Concretely, we denote states as $(x, y)$ pairs, and consider bounded 2D actions $(\delta_x, \delta_y)$ which displace the agent, i.e. $(x, y) \xrightarrow{(\delta_x, \delta_y)} (x + \delta_x, y + \delta_y)$ with

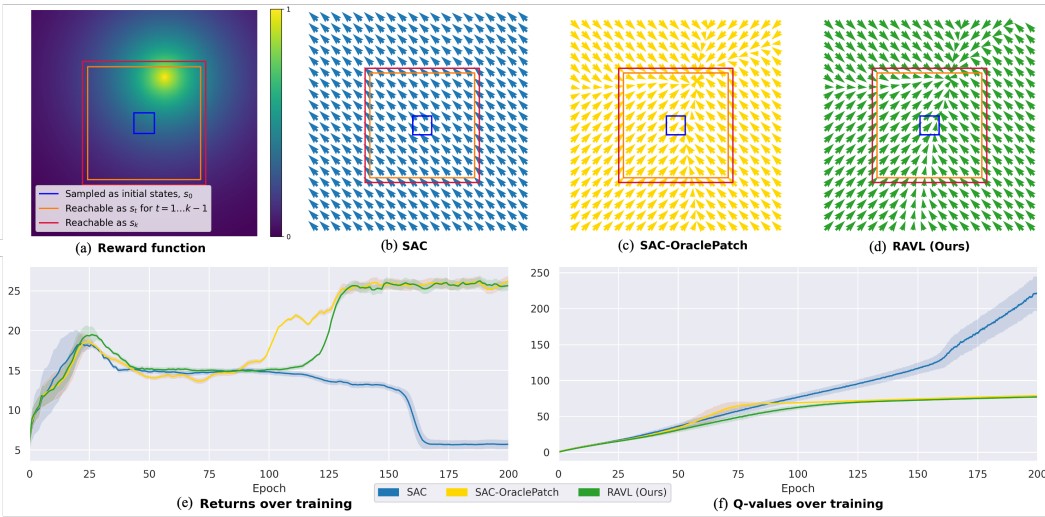

Figure 2: Experiments on the simple environment, illustrating the edge-of-reach problem and potential solutions. **(a)** Reward function, **(b)** final (failed) policy with naïve application of SAC, **(c)** final (successful) policy with patching in oracle $Q$-values for edge-of-reach states, **(d)** final (successful) policy with RAVL, **(e)** returns evaluated over training, **(f)** mean $Q$-values evaluated over training.

$|\delta_x|, |\delta_y| \leq 1$. Reward is exponentially decreasing away from the goal $g = (4, 8)$, i.e. $R(s, a) = \exp(-\frac{1}{2}\|s - g\|_2^2)$. The initial state distribution is centered around the origin, $\mu_0 = U([-2, 2]^2)$. We train the agent using SAC (Haarnoja et al., 2018) as in Sun et al. (2023); Yu et al. (2020) with a setup identical to our later D4RL benchmark experiments (see Appendix C for details). Finally, for the data collection component, the agent is allowed to sample $k = 10$ step rollouts starting from the initial state distribution $\mu_0$. We evaluate episode returns over $H = 30$ steps.

This environment serves to distill the offline RL problem into just the components relevant to the edge-of-reach issue, namely a fixed start state distribution and limited rollout horizon $k < H$. Since the actions are bounded, the set of edge-of-reach states can be defined exactly as those between the red and orange boxes in Figure 2. There is no notion of an approximate dynamics model, as we are demonstrating that issues arise even in the case of the agent having access to the true dynamics.

### 5.2 OBSERVING PATHOLOGICAL VALUE OVERESTIMATION

We may expect SAC to be able to learn a sensible policy, however, the default SAC agent (see **blue** in Figure 2) fails to train stably, with pathological $Q$-value growth over training and poor performance (Figure 2(c) and (d)). Examining the rollout trajectories sampled over training (see Figure 4) we observe the following behavior: *(Before 25 epochs)* Initially, the agent follows the reward function, and performance increases. *(Between 25 and 160 epochs)* Value misestimation takes over, and the policy begins to aim toward unobserved state-actions (since their values can be misestimated and hence overestimated), and performance subsequently decreases. *(After 160 epochs)* The effect above compounds with each epoch, leading to the agent eventually reaching edge-of-reach states. From this point onwards, the agent samples edge-of-reach states at which it never receives any corrective feedback, and the consequent pathological value overestimation results in a complete collapse in performance. As such, we see the edge-of-reach problem manifesting itself as pathological *edge-of-reach seeking behavior*. Visualizing the final policy over $\mathcal{S}$, the trained agent aims towards an arbitrary edge-of-reach state with the highest (heavily overestimated) $Q$-value, completely ignoring the reward function. We show *an analogous result holds in larger environments* in Appendix B.2.

### 5.3 VERIFYING THE HYPOTHESIS USING VALUE PATCHING

Our hypothesis is that the source of this problem is value misestimation at edge-of-reach states. We verify this with our **SAC-OraclePatch** experiments, in which we replace value estimates in the Bellman targets with their true values. In **yellow** in Figure 2 we demonstrate that applying patching solely for edge-of-reach states is sufficient to solve the problem. This is particularly compelling as in practice only a very small proportion of states had correct values patched in (0.4% of states over training). Of course, patching in true values is not possible in practice, thus in the next section we develop a practical approximation to this. We show in Section 7.1 that our proposed method, RAVL (see **green** in Figure 2), has a very similar effect to that of the ideal SAC-OraclePatch intervention and completely resolves the edge-of-reach problem.

## 6 OUR METHOD: REACH-AWARE VALUE LEARNING (RAVL)

As verified in Section 5.3, the edge-of-reach problem stems from value overestimation at edge-of-reach states. To resolve this issue, we therefore need to identify and correct values for this set of states. Edge-of-reach states can be viewed as those that may be within-distribution with respect to the set of nextstates $s'$ in $\mathcal{D}_{\text{rollouts}}$, but which are always out-of-distribution with respect to the states $s$ sampled (see Appendix A for a more detailed discussion). As such, we can use uncertainty estimation and value pessimism ideas from model-free literature (see Section 3) to identify and apply pessimism at these states (see Figure 3). Our resulting proposal is **Reach-Aware Value Learning (RAVL)**, an algorithm that unifies model-based training with Ensemble Diversified Actor-Critic (EDAC, An et al. (2021)), the current state-of-the-art model-free value pessimism approach. Concretely, RAVL involves training an ensemble of $N_{\text{critic}}$ $Q$-functions, each parameterized by $\phi_j$, and modifying Equation (2) to:

$$Q^{k+1} \leftarrow \arg\min_{Q} \mathbb{E}_{(s,a,r,s')\sim\mathcal{D},a'\sim\pi^k(\cdot|s')}[(\underbrace{Q(s,a)}_{\text{input}} - \underbrace{[r + \gamma \min_{j=1,...,N_{\text{critic}}} Q_{\phi_j}^k(s',a')]}_{\text{Bellman target}})^2] \quad (3)$$

As is standard in offline model-based RL, we optimize this objective using MBPO (Janner et al., 2019) and SAC (Haarnoja et al., 2018). We apply the ensemble diversity promoting regularizer as proposed in An et al. (2021). Full details and pseudocode are given in Appendix C. Note that, in contrast to prior model-based methods, *we do not include any dynamics uncertainty penalization*. This is discussed in Section 7.3.

The perspective taken in this paper allows us to *directly transfer ideas from the model-free literature* for dealing with the out-of-sample problem and apply them to addressing the edge-of-reach issue in the model-based setting. The resulting algorithm combines the strengths of both model-based and model-free approaches: In model-free value pessimism approaches, pessimism is required for all out-of-sample state-actions. By using the model, we are able to reduce the set of state-actions requiring pessimism to primarily those from edge-of-reach states. The resulting algorithm can hence be considered to be significantly less pessimistic.

## 7   EXPERIMENTAL RESULTS

In this section, we first evaluate RAVL on the simple environment from Section 5 and demonstrate that it provides an effective solution to the edge-of-reach problem. Next, we show that RAVL scales to strong performance on the standard D4RL benchmark. Finally, we compare RAVL to prior dynamics uncertainty penalized methods and elucidate why they may work despite not explicitly addressing the edge-of-reach problem. We provide full hyperparameters and a discussion of runtime in Appendix D.

### 7.1   EVALUATION ON THE SIMPLE ENVIRONMENT

We first test RAVL on the simple environment from Section 5 (see **green** in Figure 2). We observe that it achieves the same result as the theoretically optimal but practically impossible method of patching in the true values at edge-of-reach states, SAC-OraclePatch. The $Q$-values are stabilized over learning and match those of SAC-OraclePatch. In Figure 3, we demonstrate that RAVL is able to capture the higher value uncertainty at edge-of-reach states. The $Q$-value variance over RAVL's ensemble (RAVL's effective penalty) is significantly higher at edge-of-reach states, thus verifying that RAVL is able to detect and penalize edge-of-reach states exactly as hoped.

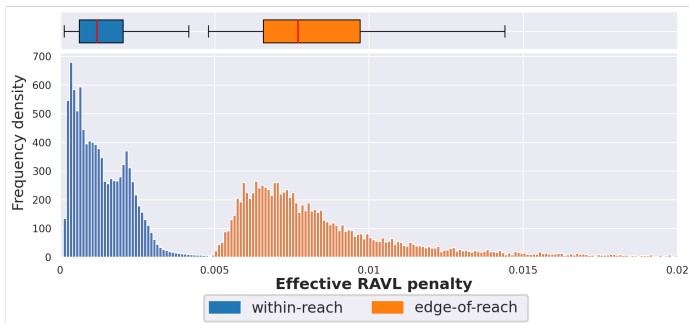

Figure 3: A histogram of RAVL's effective penalty (variance over the $Q$-value ensemble) plotted for the state-actions evaluated during training, separated into whether the state is or is not edge-of-reach. Exactly as desired, RAVL is able to detect and apply a higher penalty at edge-of-reach states. (The simple environment allows this analysis, since we can exactly define the set of edge-of-reach states.)

### 7.2   EVALUATION ON THE D4RL BENCHMARK

Next, we show that RAVL scales to the standard offline RL benchmark, D4RL (Fu et al., 2020). We consider the MuJoCo (Todorov et al., 2012) v2 datasets in Table 2 and show that we can match the performance of the current state-of-the-art in model-based method, MOBILE (Sun et al., 2023), without the need for any explicit dynamics penalization. As well as being theoretically interesting (see discussion Section 7.3), this is additionally beneficial in terms of running cost as MOBILE requires computation of multiple transition samples to calculate its penalty. Compared to EDAC (An et al., 2021), the model-free equivalent of our algorithm, RAVL achieves much higher performance on the Halfcheetah mixed and medium environments. This represents a new state-of-the-art on those datasets and shows the clear benefit of using additional synthetic model-based samples. Our approach is far stronger than COMBO (Yu et al., 2021), a competing baseline that introduced a less effective form of value conservatism into model-based methods.

Table 2: A comprehensive evaluation of RAVL over the standard D4RL MuJoCo benchmark. We show the mean and standard deviation of the final performance averaged over 6 seeds. Our simple approach matches the state-of-the-art without any explicit dynamics penalization.

| Environment | | Model-Free | | | Model-Based | | | |
|---|---|---|---|---|---|---|---|---|
| | | BC | CQL | EDAC | MOPO | COMBO | RAMBO | MOBILE | RAVL (Ours) |
| Halfcheetah | medium | 43.2 | 46.9 | 65.9 | 73.0 | 54.2 | 77.9 | 74.6 | **78.7±2.0** |
| | mixed | 37.6 | 45.3 | 61.3 | 72.1 | 55.1 | 68.7 | 71.7 | **74.9±2.0** |
| | medexp | 44.0 | 95.0 | **106.3** | 90.8 | 90.0 | 95.4 | **108.2** | 102.1±8.0 |
| Hopper | medium | 54.1 | 61.9 | 101.6 | 62.8 | 97.2 | 87.0 | **106.6** | 88.1±15.4 |
| | mixed | 16.6 | 86.3 | 101.0 | **103.5** | 89.5 | 99.5 | 103.9 | **103.1±1.0** |
| | medexp | 53.9 | 96.9 | 110.7 | 81.6 | 111.1 | 88.2 | **112.6** | 110.1±1.8 |
| Walker2d | medium | 70.9 | 79.5 | **92.5** | 84.1 | 81.9 | 84.9 | 87.7 | 86.3±1.6 |
| | mixed | 20.3 | 76.8 | 87.1 | 85.6 | 56.0 | 89.2 | **89.9** | 83.0±2.5 |
| | medexp | 90.1 | 109.1 | **114.7** | 112.9 | 103.3 | 56.7 | **115.2** | **115.5±2.4** |

## 7.3 REINTERPRETATION OF PRIOR METHODS

Finally, we compare RAVL to prior offline model-based algorithms, which overlooked the edge-of-reach problem (see Section 4) and instead primarily sought to reduce learned dynamics exploitation. We compare RAVL and MBPO (the base algorithm for most offline model-based approaches) which both **do not include any explicit dynamics penalty to prevent model exploitation**. For these algorithms, we find that the rewards collected in the model closely match, or else are lower, than in the true environment (see Tables 3 and 7). We particularly note that these levels of observed model reward could not lead to the levels of catastrophic value overestimation as seen for MBPO in Appendix B.2. This finding provides further evidence that the edge-of-reach problem may be a more accurate explanation for the issues seen in offline model-based methods.

Table 3: Per-step rewards with MBPO and RAVL in model rollouts are similar to those with the true dynamics, indicating (as in Janner et al. (2019); Lu et al. (2022)) that the model is largely accurate for short rollouts. This suggests that model exploitation is not the main issue. We provide numbers for a representative selection of D4RL environments with the configurations from Table 2. Mean and standard deviation are shown over 1000 rollouts.

| | Hopper mixed | | Walker2d medexp | | Halfcheetah medium | |
|---|---|---|---|---|---|---|
| | Model | True | Model | True | Model | True |
| **MBPO** | 2.37±1.05 | 2.44±1.02 | 3.96±1.41 | 3.96±1.41 | 4.89±1.27 | 5.01±1.09 |
| **RAVL** | 2.44±1.07 | 2.45±1.03 | 4.15±1.37 | 4.15±1.37 | 4.91±1.24 | 5.01±1.11 |

This leads us to ask: *What allows prior dynamics error-motivated methods to work despite ignoring the critical edge-of-reach problem?* To answer this, we investigate whether existing dynamics penalties may help to *accidentally mitigate* the edge-of-reach problem. We confirm this in Figure 6, showing that there is a positive correlation between the penalties in dynamics uncertainty methods and RAVL's effective penalty of value ensemble variance. This may be expected, as dynamics uncertainty will naturally be higher further away from $\mathcal{D}_{\text{offline}}$, which is also where edge-of-reach states are more likely to lie. We additionally note that, since the edge-of-reach problem is orthogonal to dynamics misestimation, RAVL could be *combined with appropriate model uncertainty penalization* (Lu et al., 2022; Sun et al., 2023; Yu et al., 2020) for environments with less accurate models.

## 8 CONCLUSION

In this paper, we develop a more complete understanding of the issues faced in offline model-based reinforcement learning. By identifying the edge-of-reach problem, we connect the previously disjoint areas of model-free and model-based offline algorithms. Our practical algorithm, RAVL, is simple and combines the strengths of both, achieving strong performance on the standard D4RL benchmark without the need for any explicit dynamics penalization. We hope the analysis in our paper inspires the future development of more principled and well-motivated offline algorithms. In this spirit, a particularly exciting future direction could be to extend the analysis in this paper to higher dimensional pixel-based settings, where model-free and model-based methods share little overlap. In particular, standard offline visual model-based algorithms such as Offline DreamerV2 (Lu et al., 2023) still rely on dynamics penalization and could be drastically simplified with our approach.

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

# SUPPLEMENTARY MATERIAL

## A  A UNIFIED PERSPECTIVE ON THE OUT-OF-SAMPLE AND EDGE-OF-REACH PROBLEMS

We supplement the discussion in Section 4.2 with a more thorough comparison of the out-of-sample and edge-of-reach problems, including how they relate to model-free and model-based approaches.

### A.1  DEFINITIONS

Consider a dataset of transition tuples $\mathcal{D} = \{(s_i, a_i, r_i, s'_i, d_i)\}_{i=1,\ldots,N}$ collected according to some dataset policy $\pi^{\mathcal{D}}(\cdot|s)$. Compared to Section 2, we include the addition of a *done* indicator $d_i$, where $d_i = 1$ indicates episode termination (and $d_i = 0$ otherwise). Transition tuples thus consist of *state*, *action*, *reward*, *nextstate*, *done*. Consider the marginal distribution over state-actions $\rho^{\mathcal{D}}_{s,a}(\cdot, \cdot)$, over states $\rho^{\mathcal{D}}_s(\cdot)$, and conditional action distribution $\rho^{\mathcal{D}}_{a|s}(\cdot|s)$. Note that $\rho^{\mathcal{D}}_{a|s}(\cdot|s) = \pi^{\mathcal{D}}(\cdot|s)$. We abbreviate *x is in distribution with respect to $\rho$* as $x \in^{\text{dist}} \rho$.

### A.2  $Q$-LEARNING CONDITIONS

As described in Section 2.1, given some policy $\pi$, we can attempt to learn the corresponding $Q$-function with the following iterative process:

$$Q^{k+1} \leftarrow \arg\min_Q \mathbb{E}_{(s,a,r,s')\sim\mathcal{D}, a'\sim\pi(\cdot|s')}[(\underbrace{Q(s,a)}_{\text{input}} - \underbrace{[r + \gamma(1-d)Q^k(s',a')]}_{\text{Bellman target}})^2] \qquad (4)$$

$Q$-learning relies on bootstrapping, hence to be successful we need to be able to learn accurate estimates of the Bellman targets for all $(s, a)$ inputs. Bootstrapped estimates of $Q(s', a')$ are used in the targets whenever $d \neq 1$. Therefore, for all $(s', a')$, we require:

> ***Combined state-action condition***: $(s', a') \in^{\text{dist}} \rho^{\mathcal{D}}_{s,a}$ or $d = 1$.

In the main paper, we use this combined state-action perspective for simplicity, however, we can equivalently divide this state-action condition into independent requirements on the state and action as follows:

> ***State condition***: $s' \in^{\text{dist}} \rho^{\mathcal{D}}_s$ or $d = 1$,
> ***Action condition***: $a' \in^{\text{dist}} \rho^{\mathcal{D}}_{a|s}(s')$ (given the above condition is met and $d \neq 1$).

Informally, the state condition may be violated if $\mathcal{D}$ consists of partial or truncated trajectories, and the action condition may be violated if there is a significant distribution shift between $\pi^{\mathcal{D}}$ and $\pi$.

### A.3  COMPARISON BETWEEN MODEL-FREE AND MODEL-BASED METHODS

Table 4: A summary of the comparison between model-free and model-based offline RL in relation to the conditions on $Q$-learning.

|  | Action condition violation | State condition violation | Main source of issues |
|---|---|---|---|
| **Model-free** | Possible, common in practice | Possible, but uncommon in practice | → Action condition violation |
| **Model-based** | Not possible | Possible, common in practice | → State condition violation |

In offline model-free RL, $\mathcal{D} = \mathcal{D}_{\text{offline}}$, with $\pi^{\mathcal{D}} = \pi^\beta$. For the settings we consider, $\mathcal{D}_{\text{offline}}$ consists of full trajectories and therefore will not violate the state condition. However, this may happen in a more general setting with $\mathcal{D}_{\text{offline}}$ containing truncated trajectories. By contrast, the mismatch between $\pi$ (used to sample $a'$ in $Q$-learning) and $\pi^\beta$ (used to sample $a$ in the dataset $\mathcal{D}_{\text{offline}}$) often does lead to **significant violation of the action condition**. This exacerbates the overestimation bias in $Q$-learning (see Section 3), and can result in pathological training dynamics and $Q$-value explosion over training (Kumar et al., 2019).

On the other hand, in offline model-based RL, the dataset $\mathcal{D} = \mathcal{D}_{\text{rollouts}}$ is collected *on-policy* according to the current (or recent) policy such that $\pi^{\mathcal{D}} \approx \pi$. This minimal mismatch between $\pi^{\mathcal{D}}$ and $\pi$ means the action condition is not violated and can be considered to be resolved due to the collection of additional data. However, the procedure of generating the data $\mathcal{D} = \mathcal{D}_{\text{rollouts}}$ can be seen to significantly exacerbate the state condition problem, as the use of short truncated-horizon trajectories means the resulting dataset $\mathcal{D}_{\text{rollouts}}$ is **likely to violate the state condition**. Due to lack of exploration, certain states may temporarily violate the state condition. Our paper then considers the pathological case of *edge-of-reach* states, which will always violate the state condition.

## B   ADDITIONAL FIGURES

### B.1   VISUALIZATIONS OF ROLLOUTS THROUGH TRAINING

In Figure 4, we provide a visualization of the rollouts sampled over training in the simple environment for each of the algorithms analyzed in Figure 2 (see Section 5). This accompanies the discussion of the behavior of SAC during training in Section 5.2.

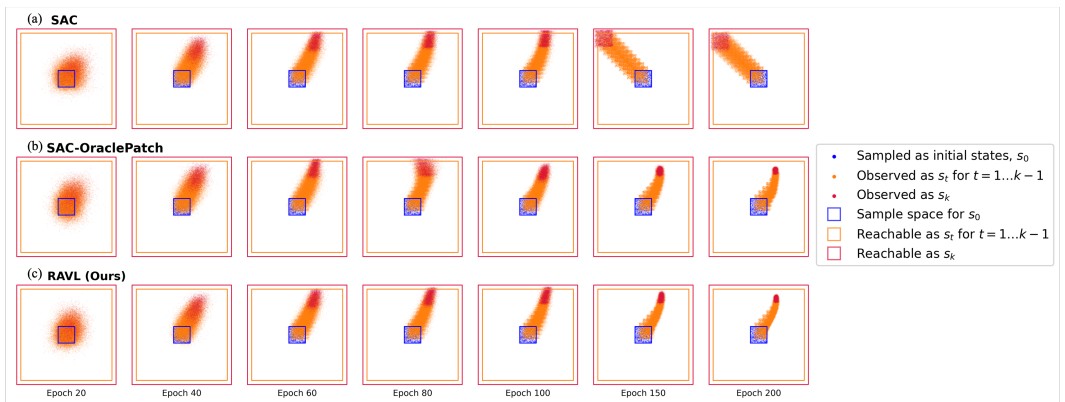

Figure 4: A visualization of the rollouts sampled over training on the simple environment in Section 5. We note the pathological behavior of the baseline SAC, and the success of the ideal intervention SAC-OraclePatch, and our practically realizable method RAVL.

### B.2   OBSERVING PATHOLOGICAL VALUE OVERESTIMATION ON THE D4RL BENCHMARK

In Figure 5, we show observations of pathological overestimation of $Q$-values on a representative D4RL dataset when optimizing using MBPO (which corresponds to $N_{\text{critic}} = 2$ and $\eta = 0$ in our method). This confirms that despite the ability to gather additional on-policy data, the out-of-sample issue effectively persists in model-based methods. RAVL is able to resolve this issue and achieve state-of-the-art performance.

### B.3   COMPARISON BETWEEN DYNAMICS PENALTY AND RAVL

We provide a further comparison to Section 7.3 between our method and prior offline model-based methods by comparing the dynamics uncertainty-based penalty used in MOPO with the variance of the value ensemble of RAVL in Figure 6. We find there is a positive correlation between the dynamics uncertainty-based penalty used in MOPO (Yu et al., 2020), and the variance of the value ensemble of RAVL. This helps us in reinterpreting dynamics uncertainty-based methods and explaining why they may work despite not considering the crucial edge-of-reach problem.

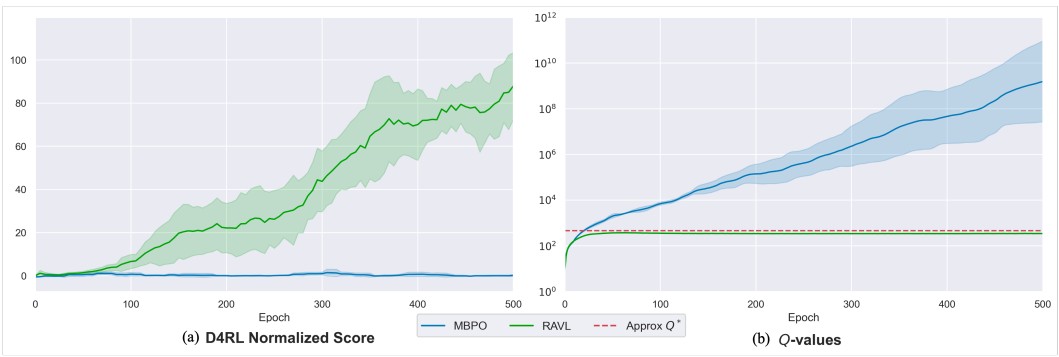

Figure 5: We observe an exponential increase in $Q$-values and resulting low performance on one of the D4RL datasets using the baseline MBPO. We present results on Walker2d-medexp but note that similar trends are seen across other D4RL datasets. RAVL is able to resolve these issues and achieve state-of-the-art performance. This is a similar trend to that demonstrated in our simple environment in Figure 2(f). The figure shows **(a)** D4RL normalized score and **(b)** mean $Q$-values evaluated over training. *Approx $Q^*$* indicates the magnitude of $Q$-values which would correspond to a normalized D4RL score of 100 with $\gamma = 0.99$. We show the mean and standard deviation averaged over 6 seeds.

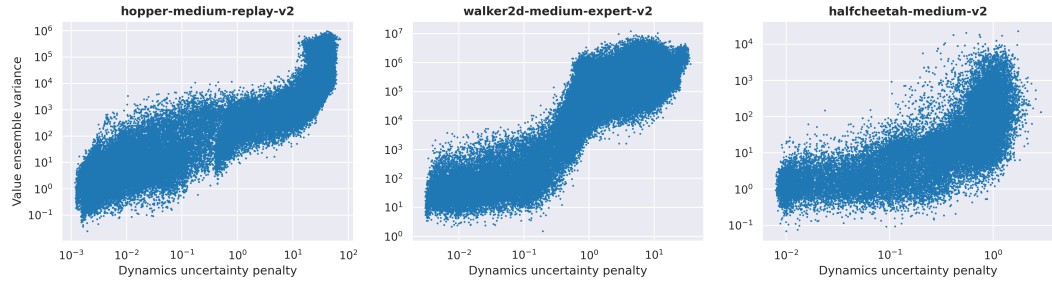

Figure 6: We find that the dynamics uncertainty-based penalty used in MOPO (Yu et al., 2020) is positively correlated with the variance of the value ensemble of RAVL, suggesting prior methods may unintentionally address the edge-of-reach problem. Pearson correlation coefficients are 0.49, 0.43, and 0.27 for Hopper-mixed, Walker2d-medexp, and Halfcheetah-medium respectively.

## C  IMPLEMENTATION DETAILS

In this section, we give full implementation details for RAVL. As is typical in offline model-based reinforcement learning (Kidambi et al., 2020; Lu et al., 2022; Sun et al., 2023; Yu et al., 2020), we use short MBPO (Janner et al., 2019) style rollouts. The agent is trained using the EDAC (An et al., 2021) losses, a conservative ensemble-based algorithm based on SAC (Haarnoja et al., 2018). Our transition model follows the standard setup in model-based offline algorithms, being realized as a deep ensemble (Chua et al., 2018) and trained via maximum likelihood estimation. We summarize this by providing pseudocode for offline model-based methods. The pseudocode representing prior dynamics uncertainty-based methods is given in Algorithm 1 for comparison, while RAVL is represented in Algorithm 2.

## D  HYPERPARAMETERS

For the D4RL (Fu et al., 2020) MuJoCo datasets presented in Section 7.2, we sweep over the following hyperparameters and list the optimal choices in Table 5.

- **(EDAC)** Number of $Q$-ensemble elements $N_{\text{critic}}$, in the range $\{10, 50\}$
- **(EDAC)** Ensemble diversity weight $\eta$, in the range $\{1, 10, 100\}$
- **(MBPO)** Model rollout length $k$, in the range $\{1, 5\}$

---

**Algorithm 1** Pseudocode for base model-based algorithm (MBPO)

---

1: **Require:** $\widehat{M} = (\widehat{T}, \widehat{R})$ learned environment model (trained on $\mathcal{D}_{\text{offline}}$)
        (Existing methods include dynamics uncertainty penalty in $\widehat{M}$)
2: **Input:** Rollout length $k \geq 1$, real data ratio $r \in [0, 1]$, $\mathcal{D}_{\text{offline}}$ offline dataset
3: **Initialize:** $\mathcal{D}_{\text{rollouts}} = \emptyset$ model replay buffer, $\pi_\theta$ policy, $Q_\phi$ value ensemble
4: **for** epochs $= 1, \ldots$ **do**
5:     **(Collect data)** Starting from states in $\mathcal{D}_{\text{offline}}$, collect $k$-step rollouts in $\widehat{T}$ with $\pi_\theta$. Store data in $\mathcal{D}_{\text{rollouts}}$
6:     **(Train agent)** Train $\pi_\theta$ and $Q_\phi$ on $\mathcal{D}_{\text{rollouts}} \cup \mathcal{D}_{\text{offline}}$ mixed with ratio $r$
7: **end for**

---

**Algorithm 2** Pseudocode for RAVL

---

1: **Require:** $\widehat{M} = (\widehat{T}, \widehat{R})$ learned environment model (trained on $\mathcal{D}_{\text{offline}}$)
2: **Input:** Rollout length $k \geq 1$, real data ratio $r \in [0, 1]$, $\mathcal{D}_{\text{offline}}$ offline dataset
3: **Initialize:** $\mathcal{D}_{\text{rollouts}} = \emptyset$ model replay buffer, $\pi_\theta$ policy, $Q_\phi$ value ensemble
4: **for** epochs $= 1, \ldots$ **do**
5:     **(Collect data)** Starting from states in $\mathcal{D}_{\text{offline}}$, collect $k$-step rollouts in $\widehat{T}$ with $\pi_\theta$. Store data in $\mathcal{D}_{\text{rollouts}}$
6:     **(Train agent)** Train $\pi_\theta$ and $Q_\phi$ on $\mathcal{D}_{\text{rollouts}} \cup \mathcal{D}_{\text{offline}}$ mixed with ratio $r$ using EDAC for $Q$-ensemble
7: **end for**

---

- **(MBPO)** Real-to-synthetic data ratio $r$, in the range $\{0.05, 0.5\}$

The remaining model-based and agent hyperparameters are given in Table 6. We note that, similarly to An et al. (2021), the Hopper environment performs better with $N_{\text{critic}} = 50$ while Walker2d and Halfcheetah perform better with $N_{\text{critic}} = 10$; thus in practice, **we only need to tune the remaining three hyperparameters**. Our implementation is based on the Clean Offline Reinforcement Learning (CORL, Tarasov et al. (2022)) repository, released at `https://github.com/tinkoff-ai/CORL` under an Apache-2.0 license. Our algorithm takes approximately 1 day to run on the standard D4RL datasets using an A100 GPU for the full number of epochs.

Table 5: Variable hyperparameters for RAVL used in D4RL MuJoCo locomotion tasks.

| Environment | | $N_{\text{critic}}$ | $\eta$ | k | r |
|---|---|---|---|---|---|
| | medium | 10 | 1 | 5 | 0.05 |
| HalfCheetah | mixed | 10 | 100 | 5 | 0.05 |
| | medexp | 10 | 1 | 5 | 0.5 |
| | medium | 50 | 100 | 1 | 0.5 |
| Hopper | mixed | 50 | 10 | 1 | 0.5 |
| | medexp | 50 | 100 | 1 | 0.5 |
| | medium | 10 | 10 | 1 | 0.5 |
| Walker2d | mixed | 10 | 1 | 5 | 0.05 |
| | medexp | 10 | 1 | 1 | 0.5 |

Table 6: Fixed hyperparameters for RAVL used in D4RL MuJoCo locomotion tasks.

| Parameter | Value |
|---|---|
| epochs | 3,000 |
| gamma | 0.99 |
| learning rate | $3 \times 10^{-4}$ |
| batch size | 256 |
| buffer retain epochs | 5 |
| number of rollouts | 50,000 |

# E   Proof of Error Propagation Result

In this section, we provide a proof of Proposition 1. Our proof follows analogous logic to the error propagation result of Kumar et al. (2019).

**Proposition 1.** *[Error propagation from edge-of-reach states] Consider a rollout of length $k$, $(s_0, a_0, s_1, \ldots, s_k)$. Suppose that the state $s_k$ is edge-of-reach and the approximate value function $Q^{j-1}(s_k, \pi(s_k))$ has error $\epsilon$. Then, standard value iteration will compound error $\gamma^{k-t}\epsilon$ to the estimates of $Q^j(s_t, a_t)$ for $t = 1, \ldots, k-1$. Proof provided in Appendix E.*

*Proof.* Let us denote $Q^*$ as the optimal value function, $\zeta_j(s, a) = |Q_j(s, a) - Q^*(s, a)|$ the error at iteration $j$ of Q-Learning, and $\delta_j(s, a) = |Q_j(s, a) - \mathcal{T}Q_{j-1}(s, a)|$ the current Bellman error. Then first considering the $t = k - 1$ case,

$$\zeta_j(s_t, a_t) = |Q_j(s_t, a_t) - Q^*(s_t, a_t)| \tag{5}$$
$$= |Q_j(s_t, a_t) - \mathcal{T}Q_{j-1}(s_t, a_t) + \mathcal{T}Q_{j-1}(s_t, a_t) - Q^*(s_t, a_t)| \tag{6}$$
$$\leq |Q_j(s_t, a_t) - \mathcal{T}Q_{j-1}(s_t, a_t)| + |\mathcal{T}Q_{j-1}(s_t, a_t) - Q^*(s_t, a_t)| \tag{7}$$
$$= \delta_j(s_t, a_t) + \gamma\zeta_{j-1}(s_{t+1}, a_{t+1}) \tag{8}$$
$$= \delta_j(s_t, a_t) + \gamma\epsilon \tag{9}$$

Thus the errors at edge-of-reach states are discounted and then compounded with new errors at $Q^j(s_{k-1}, a_{k-1})$. For $t < k - 1$, the result follows from repeated application of Equation (8) along the rollout. $\square$

## F    ADDITIONAL TABLES

### F.1    FULL HORIZON MODEL ROLLOUTS

Complementary to Table 3, we show that returns for full horizon model rollouts ($k = 1000$) are lower than the true returns in Table 7. This provides further evidence that model exploitation may not be an issue even for long horizons. However, we note that learned dynamics models are not expected to be accurate to this horizon, so returns may be arbitrarily inaccurate.

Table 7: Returns of MBPO and RAVL evaluated in the learned dynamics and reward model used for training are always lower than the true returns, suggesting that model exploitation is not the main issue. We provide numbers for a representative selection of D4RL environments with the configurations from Table 2. Normalized mean and standard deviation are shown over 6 random seeds.

|  | Hopper mixed | | Walker2d medexp | | Halfcheetah medium | |
|---|---|---|---|---|---|---|
|  | **Model** | **True** | **Model** | **True** | **Model** | **True** |
| **MBPO** | $47.5\pm15.3$ | $71.5\pm32.2$ | $-0.2\pm-0.4$ | $7.7\pm2.1$ | $-501.6\pm45.6$ | $72.9\pm8.6$ |
| **RAVL** | $89.3\pm7.3$ | $103.0\pm0.7$ | $18.9\pm7.1$ | $115.5\pm2.4$ | $40.2\pm42.6$ | $78.7\pm2.0$ |

### F.2    SUMMARY OF SETUPS USED IN COMPARISONS

Throughout the paper, we compare several different setups in order to identify the true underlying issues in model-based offline RL. We provide a summary of them in Table 8. More comprehensive descriptions of each are given in the main text and in relevant table and figure captions.

Table 8: We summarize the various setups used for comparisons throughout the paper. '*' denotes application to the simple environment (see Section 5). All methods use $k$-step rollouts from the offline dataset (or from a fixed starting state distribution in the case of the simple environment).

|  | **Dynamics Model** | | **Agent** | | **Figures and Tables** |
|---|---|---|---|---|---|
|  | Type | Penalty | $N_{\text{critic}}$ | $\eta$ | |
| **MOPO** | Ensemble | ✓ | 2 | 0 | *Table* 1, *Figure* 6 |
| **Oracle** | True | n/a | 2 | 0 | *Table* 1 |
| **MBPO (Offline)** | Ensemble | ✗ | 2 | 0 | *Tables* 3, 7, *Figure* 5 |
| **RAVL (Ours)** | Ensemble | ✗ | $> 2$ | $> 0$ | *Tables* 2, 3, 7, *Figure* 5 |
| **SAC (Truncated Horizon)*** | True | n/a | 2 | 0 | *Figures* 2, 4 |
| **RAVL (Ours)*** | True | n/a | $> 2$ | $> 0$ | *Figures* 2, 3, 4 |

