# OpenReview forum: "RAVL: Reach-Aware Value Learning for the Edge-of-Reach Problem in Offline Model-Based Reinforcement Learning"
_ICLR.cc/2024/Conference — Submitted to ICLR 2024_

### Official Review · Reviewer_34p6 · 2023-10-25

**Soundness:** 2 fair
**Presentation:** 3 good
**Contribution:** 2 fair
**Rating:** 3
**Confidence:** 3

**Summary:**

This paper studies offline MBRL. The authors first show a surprising experimental result: the existing offline MBRL method MOPO does not work when replacing learned models with true dynamics models. They claim that the reason for this failure is the overestimation of values on the edge-of-reach states which are only reached at the final time steps of the limited horizon rollouts. To address this issue, they propose a new method RAVL which combines EDAC and MOPO. They validate the performance of RAVL on a simple 2D environment and D4RL benchmarks.

**Strengths:**

1. The experiments in Section 4.1 are very interesting and insightful. They found that MOPO surprisingly fails when replacing learned models with true dynamics models. This result provides a new understanding of offline MBRL: the model error is not the only issue in offline MBRL.
2. The paper is well-written and easy to follow, providing clear explanations and detailed descriptions of the proposed method and experimental results.

**Weaknesses:**

1. The authors first reveal the key overestimation issue of values on edge-of-reach states in offline MBRL. However, as a direct combination of EDAC and MOPO, the proposed method RAVL is largely independent of the revealed issue. In particular, the only difference between RAVL and MOPO is that RAVL exhibits the Q-update rule in EDAC. However, such a Q-update rule is performed on all states in the buffer and is not tailored for edge-of-reach states. RAVL does not identify edge-of-reach states and correct the corresponding values to address the claimed issue.
2. There is a large gap between the formalization in Section 4.3 and actual offline MBRL methods. First, the definition of edge-of-reach states is very limited. Concretely, such a definition only considers the case where the starting states of model rollouts are only sampled from the initial state distribution. However, in offline MBRL,  the starting state could be any state along the trajectories in the offline dataset. Such a gap leads to a problem that the Edge-of-reach states defined in Definition 1 could be **not** edge-of-reach in model rollouts with different starting states. For instance, let $(s_0, a_0, s_1, a_1, s_2, a_2)$ be a trajectory in the offline dataset. We consider two types of model rollouts with different starting states: $(s_0, \hat{a}_0, \hat{s}_1, \hat{a}_1, \hat{s}_2, \hat{a}_2)$ and $(s_1, \tilde{a}_1, \tilde{s}_2, \tilde{a}_2,  \tilde{s}_3, \tilde{a}_3)$. Here $\hat{s}_2$ is edge-of-reach for the first type of model rollouts but could be not edge-of-reach for the second type of model rollouts.
Second, Proposition 1 considers an extremely simple case where the Q-update is performed only on a single model rollout. However, in MBRL, Q-updates are performed on multiple model rollouts where states could be overlapped. In this case, the analysis of Q-functions could be much more complicated.

3. The empirical performance of RAVL is not strong. In D4RL, the existing method MOBILE beats RAVL regarding the number of best-performance tasks, implying that the improvement of RAVL is limited.

**Questions:**

Typos:

1. In Eq.(3), $+\gamma$ should be $-\gamma$

---

> ### Author Response · Authors · 2023-11-15
> **Response to Reviewer 34p6**
>
> We would like to thank the reviewer for their time. We are pleased that you found our experiments “interesting and insightful” and that the paper was “well-written and easy to follow”.
>
> **(Weakness 1) Does RAVL actually target edge-of-reach states?**
>
> This is a good question and gets to the crux of our method. Please see part A of our general response where we have included a detailed step-by-step walk-through of the intuition of why we can expect our RAVL approach to detect and penalize edge-of-reach states. We would be more than happy to provide further explanation, so please let us know if it is still unclear.
>
> **(Weakness 2, part 1) What starting states are being sampled in Definition 1?**
>
> Thank you for raising this. In hindsight, the notation in Definition 1 is slightly confusing. In Definition 1 we used $\mu_0$ to denote the starting state distribution of the rollouts (i.e. the distribution of $\mathcal{D}_{\text{offline}}$) as opposed to the initial state distribution of the environment. Thus, the edge-of-reach definition here is the correct one for offline model-based RL, and matches the one you describe. We apologize for causing confusion here and have adjusted the notation in Definition 1 to make this clearer.
>
> **(Weakness 2, part 2) Analysis of the Q-function update**
>
> Thank you for raising this. We provide this analysis as an illustrative example of how errors can be propagated from edge-of-reach states. We agree with the reviewer that in practice, model rollouts are appended to a buffer and training is performed on mini-batches of this data. However, as demonstrated empirically, the same optimization dynamics are likely to occur over the course of training, with edge-of-reach states not receiving any updates and errors from them propagating to all states. We have further clarified this in the paper after Propositon 1.
>
> **(Weakness 3) Empirical performance of RAVL**
>
> Thank you for raising this comment, however, we respectfully disagree with the assessment that the performance of RAVL is not strong, as it is state-of-the-art in 5/9 of the environments we evaluated on. In addition, we are pleased to be able to report that RAVL gives significant performance boosts on the more complex V-D4RL benchmark *(please see part B of our general response)*. We also note that SOTA scores on the D4RL benchmark are likely saturated, already being significantly over 100 (normalized expert level) on many of the environments. As such, future work is unlikely to be able to demonstrate substantial improvements on this benchmark.
>
> More importantly, however, we have demonstrated that MOBILE (the only algorithm that beats RAVL), along with many other prominent offline model-based algorithms, is derived based on significantly incomplete theoretical foundations (and we explain why they work despite this). It is the new insight, and the resulting much more complete understanding of offline RL, that we view as the main exciting new contribution of our work. We anticipate this should be hugely valuable in the future development of robust, strong-performing offline RL methods, potentially also resulting in new improved ways of tackling the critical, previously overlooked edge-of-reach problem.
>
> **(Question 1) Typo in Equation 3**
>
> Thank you for spotting the typo in Equation 3. We have now fixed this in the updated paper.
>
> ---
>
> Thank you again for your insightful comments and questions. Spotting the typo and your feedback on the notation in definition 1 were especially helpful in improving the paper. We hope our responses were able to address any concerns and questions you had and convince you of the contribution of our work. Please let us know if you have any remaining concerns or if anything is still unclear - we would be more than happy to discuss further. If we have been able to address your concerns, we humbly ask if you would consider raising your score.

---

> ### Author Response · Authors · 2023-11-21
>
> Dear Reviewer 34p6,
>
> Thank you again for taking the time to review our paper. As author-reviewer discussion period is about to end (22nd Nov AOE, approximately two days), we were wondering whether our clarifications were able to address your questions and clear up any misunderstandings about our algorithm and the formalization in the paper.
>
> Please let us know if you have any remaining questions or concerns. We hope that we have addressed your queries, and if so, would appreciate it if you would consider revising your score.
>
> Best wishes,
>
> The authors.

---

> > ### Comment · Reviewer_34p6 · 2023-11-23
> >
> > Thank the authors for the detailed response!
> >
> > However, my major concern about how RAVL identifies and handles edge-of-reach states still remains. In the authors’ response, they explained that RAVL identifies edge-of-reach states by detecting high-uncertainty states (OOD states). However, the set of edge-of-reach states is only a strict subset of high-uncertainty states since high-uncertainty states also contain states that could be visited in the middle of trajectories but are uncovered in the offline dataset. Moreover, previous offline model-based and model-free methods have proposed to identify high-uncertainty states using model uncertainty and Q-uncertainty, respectively. Thus, it is difficult to discern the distinct advantage of RAVL over existing methods from the perspective of edge-of-reach states.

---

> ### Author Response · Authors · 2023-11-23
> **Further clarification**
>
> Dear Reviewer 34p6,
>
> The OOD (high uncertainty) states that RAVL detects are **with respect to $D_{\text{rollouts}}$ and not $D_{\text{offline}}$**. Therefore, the scenario the reviewer describes could not happen and RAVL specifically targets edge of reach states which are by definition OOD with respect to $s\in D_{\text{rollouts}}$. This allows us to use existing Q-uncertainty methods but **with respect to the model rollout distribution and not the original offline data distribution**, and is the key novelty of our method.
>
> Please let us know if we can help clarify further.
>
> Best,
> The authors.

---

### Official Review · Reviewer_RQph · 2023-10-31

**Soundness:** 3 good
**Presentation:** 3 good
**Contribution:** 3 good
**Rating:** 6
**Confidence:** 3

**Summary:**

This paper presents an intriguing finding that challenges the conventional view of model-based offline reinforcement learning: the erroneous dynamical model does not account for the behavior of model-based methods; rather, it is the overestimation of the value of states that are difficult to reach that explains their behavior. This paper proposes a simple Reach-Aware Value Learning to solve the out-of-sample problem by capturing value uncertainty at edge-of-reach states. The illustration experiment is sound and makes sense.

**Strengths:**

+ This paper is meticulously written and excellently structured, providing ample background information and a well-defined problem statement.

+ The findings presented in this paper are interesting, revealing how the overestimation of the value of states that are difficult to reach can have a significant impact on the optimization of offline model-based RL policies.

+ The proposed solution, RAVL, is both simple to implement and highly effective in addressing this problem.

**Weaknesses:**

+ **The problem addressed in this paper may not be influential.**
+ **The benchmark environment used in this research is relatively simplistic.** It would be interesting to investigate whether or not the proposed method performs satisfactorily in more complex environments.
+ **The modifications made to the experiments in this paper do not appear to be particularly substantial.**

**Questions:**

+ I do not fully understand the Figure 3. How is the figure generated? Why is the conclusion drawn: "As desired, RAVL is effective at capturing the value uncertainty for state-actions which transition to edge-of-reach nextstates."?

---

> ### Author Response · Authors · 2023-11-15
> **Response to Reviewer RQph**
>
> We would like to thank the reviewer for their time in reviewing our submission. We are pleased that you found that our paper presented an “intriguing finding” and are encouraged by your comment of it being “meticulously written and excellently structured.”
>
> **(Weakness 1) Influence of the paper**
>
> Thank you for this question. We strongly believe *(like reviewer y8bf)* that the analysis in our paper has the potential to have a substantial impact on future work in this area.
>
> The main contribution of our paper is to expose and explain a significant gap in the current understanding of the issues underlying offline model-based RL. This current incomplete *“dynamics model error”*-based view forms the basis for many current offline model-based algorithms, such as MOPO [1], MOReL [2], and MOBILE [3]. Combined, these works alone have a total of over 1000 citations and have influenced a whole sub-field of offline model-based RL. Our paper demonstrates a significant misunderstanding in these works, provides and verifies an explanation (the edge-of-reach problem), as well as providing a solution (RAVL). We believe the resulting much more complete picture of offline model-based RL should be hugely valuable in the development of more accurately motivated and stronger-performing offline algorithms.
>
> Furthermore, our detailed experiments show the edge-of-reach problem is not just an issue in theory, but that it is actually very prevalent in practice (on the simple environment, on the standard offline D4RL benchmark (see Table 1 and Figure 5), and even in latent space models *(see our new results in part B of our general response*).
>
> Finally, as discussed in Section 6 and Appendix A, our work provides a unified view of model-based and model-free approaches, which we hope will enable more ideas from model-free literature to be directly applied to model-based algorithms (as we have done with RAVL), and we anticipate that our work may inspire further future connections to be drawn between the two subfields of offline RL.
>
> **(Weakness 2) Evaluation on harder environments**
>
> Thank you for raising this concern. We are pleased to present additional state-of-the-art results on the more challenging pixel-based V-D4RL benchmark *(please see part B of our general response)*.
>
> **(Weakness 3) Simplicity of our algorithm**
>
> Thank you for noting that the approach in RAVL is quite simple. We agree, and we see this as a strength. Our aim was to make as few modifications to the base algorithm as possible, and our unified view of model-based and model-free approaches (introduced in Section 6 and Appendix A) enables us to directly take a robust out-of-the-box solution from model-free literature and apply it to solve the edge-of-reach problem. We found this worked extremely well and found no need to add anything to complicate it.
>
> **(Question 1) How was Figure 3 generated?**
>
> Thank you for raising this question. It has brought to our attention that the caption and labels in Figure 3 were unnecessarily unclear and we have edited them in light of your comment. In short: our construction of the simple environment allows us to exactly define whether a state is edge-of-reach or not (checking whether it is within fixed bounds). We are therefore able to compare the Q-ensemble variance for edge-of-reach vs. not edge-of-reach states. Figure 3 shows a histogram of the Q-ensemble variance (RAVL’s effective penalty) for all the state-action pairs evaluated over training, separated into whether the state is edge-of-reach (orange) or not edge-of-reach (ie. within-reach, blue). It shows that, as we hoped, the Q-ensemble variance (RAVL’s effective penalty) is significantly higher on states which are edge-of-reach, thus verifying that our RAVL method is effective at detecting and penalizing edge-of-reach states. (In part A of our general response we reiterate the intuition on how RAVL is able to do this.)
>
> ---
>
> We would once again like to thank you for providing insightful feedback on our submission that helped us improve our paper. We hope our responses have clarified any questions you may have had. Please let us know if you have any remaining concerns or queries as we would be more than happy to discuss further. If we have been able to address your concerns, we humbly ask if you would consider raising your score.
>
> ---
>
> [1] MOPO: Model-based Offline Policy Optimization Tianhe Yu, Garrett Thomas, Lantao Yu, Stefano Ermon, James Zou, Sergey Levine, Chelsea Finn, Tengyu Ma. NeurIPS, 2020.
>
> [2] MOReL: Model-Based Offline Reinforcement Learning Rahul Kidambi, Aravind Rajeswaran, Praneeth Netrapalli, Thorsten Joachims. NeurIPS, 2020.
>
> [3] Model-Bellman Inconsistency for Model-based Offline Reinforcement Learning Yihao Sun, Jiaji Zhang, Chengxing Jia, Haoxin Lin, Junyin Ye, Yang Yu. ICML, 2023.

---

> ### Comment · Reviewer_RQph · 2023-11-21
>
> Thank you for the author's response, which basically solves my questions. Therefore, I tend to recommend this paper to the ICLR community. However, the "highlight" may be a bit too strong for this paper, so I prefer to keep the original rating (6).

---

> > ### Author Response · Authors · 2023-11-21
> >
> > Dear Reviewer RQph,
> >
> > Thank you again for taking the time to review our paper. We are glad that our response clarified your questions and that you agreed to recommend our work for ICLR!
> >
> > Best wishes,
> >
> > The authors.

---

### Official Review · Reviewer_y8bf · 2023-10-31

**Soundness:** 4 excellent
**Presentation:** 4 excellent
**Contribution:** 4 excellent
**Rating:** 8
**Confidence:** 5

**Summary:**

The paper begins by presenting the surprising finding that typical model-based offline RL algorithms fail when provided with oracle dynamics, flaunting the conventional wisdom that these algorithms aim primarily to address exploitation of model inaccuracies. This finding leads to the main conceptual contribution of the paper, which is the “edge of reach” problem: in these model-based offline RL algorithms, which use short-horizon rollouts to collect additional synthetic data for training, some states can only be reached in the final step of rollouts, and thus the Bellman backup targets computed at those states are prone to estimation error. Based on this understanding, the authors propose Reach-Aware Value Learning (RAVL), which eschews explicit dynamics uncertainty quantification in favor of pessimism with respect to a critic ensemble, as in Ensemble-Diversified Actor Critic (EDAC), a model-free offline RL algorithm. RAVL is evaluated on a subset of the D4RL benchmark, where it exhibits performance competitive with the SOTA.

**Strengths:**

* The paper demonstrates a significant misconception in the literature of model-based offline RL, i.e. that addressing model inaccuracy is a primary reason why these algorithms can succeed. This finding, along with the identification of the edge-of-reach problem, is likely to have a substantial impact on future algorithmic work in this area.
* In addition to evaluating on a subset of the standard D4RL benchmark, the authors include a more in-depth exploration in a simple environment where the edge-of-reach issue can be cleanly studied.
* The proposed algorithm, RAVL, addresses the edge-of-reach problem and displays compelling performance without explicitly quantifying dynamics uncertainty (which is a challenging problem).

**Weaknesses:**

The D4RL evaluation includes only the basic MuJoCo tasks. The authors could try a more complex environment such as AntMaze or Adroit to further demonstrate the strength of the algorithm over previous methods.

**Questions:**

While RAVL seems to be effective at addressing the edge-of-reach problem, it may require a large ensemble (e.g. 50 models) to achieve sufficient pessimism, which could be computationally expensive. I was curious if you considered/experimented with adding a small explicit penalty to the values of states at the final step of the rollouts?

---

> ### Author Response · Authors · 2023-11-15
> **Response to Reviewer y8bf**
>
> We would like to thank the reviewer for their time in reviewing our submission. We really appreciate that you have taken the time to understand what we are trying to show, and are encouraged that, like us, you believe this work “is likely to have a substantial impact on future algorithmic work in this area”.
>
> **(Weakness 1) Evaluation on harder environments**
>
> Thank you for raising this concern. We have run RAVL on the more challenging pixel-based V-D4RL benchmark and are pleased to report new state-of-the art results *(please see Part B of our general response)*.
>
> **(Question 1, part 1) Can the number of critics be reduced?**
>
> In our original submission, we stated that RAVL required N=50 critics to achieve sufficient edge-of-reach awareness on the hopper environments (while N=10 was enough for all the other environments). We are pleased to report that we have run additional experiments and have found that N=10 is mostly sufficient for hopper as well, with only hopper-medexp needing N=30, meaning that all but one environment uses N=10.
>
> | **Environment** |        | **RAVL (N=50, in the paper)** |  **RAVL (Updated)**  |
> |:---------------:|:------:|:-----------------------------:|:--------------------:|
> | hopper-v2       | mixed  | 103.1±1.0                     | 102.6±1.1 **(N=10)** |
> | hopper-v2       | med    | 88.1±15.4                     | 90.6±11.9 **(N=10)** |
> | hopper-v2       | medexp | 110.1±1.8                     | 107.0±5.4 **(N=30)** |
>
> **(Question 1, part 2) Did you try adding an explicit penalty to states at the end of rollouts as an alternative approach to the edge-of-reach problem?**
>
> Yes, this was an idea we had initially as well! We considered this for quite some time, but from our experiments it did not seem to work well. Our intuition on why this may not work is: Final step rollout states may not actually be edge-of-reach and hence may also appear earlier in a different rollout. This means the target for the state will sometimes have a penalty and sometimes have no penalty. From our investigations, it seems that the resulting contradictory target function or reward signal leads to significant optimization issues. As such there is an extremely delicate trade-off between balancing the penalty to be (1) large enough to address the edge-of-reach issue, but (2) small enough to not cause optimization issues. This highlights the challenge of tackling the edge-of-reach problem! We found our RAVL approach to work much better and more robustly and avoid this optimization difficulty and delicate balancing act.
>
> ---
>
> We would again like to thank you for taking the time to understand and review our work. We especially appreciated your insightful suggestions and encouraging feedback. Please let us know if you have any other comments, concerns, or questions.

---

> > ### Comment · Reviewer_y8bf · 2023-11-22
> > **Response to authors**
> >
> > Thank you for your response and the additional experiments! For the pixel-based tasks, I think it is hard to argue that the improvement is "significant" in a statistical sense (considering the high standard deviation), but RAVL does improve the mean and tends to reduce the variance so I think it is still a useful addition. It is also good to see that one can use only 10 critics in most cases.
> >
> > I would say that my concerns are addressed and still recommend acceptance.

---

> > > ### Author Response · Authors · 2023-11-22
> > > **Thank you for the response**
> > >
> > > Dear Reviewer y8bf,
> > >
> > > Thank you again for taking the time to review our paper. We are glad that our response clarified your questions and that you recommend acceptance for our work at ICLR! We will include the pixel-based experiments in our camera-ready paper with this discussion.
> > >
> > > Best wishes,
> > >
> > > The authors.

---

### Official Review · Reviewer_1Cj8 · 2023-11-01

**Soundness:** 2 fair
**Presentation:** 3 good
**Contribution:** 2 fair
**Rating:** 3
**Confidence:** 3

**Summary:**

This paper focuses on the model-based offline reinforcement leanring algorithms, and different with prior works, this paper finds that the current model-based offline RL algorithms still fail even if the real dynamic model is accessible and attributes this failure to the edge-of-reach problem. Then the authors propose Reach-Aware Value Learning (RAVL) to mitigate this new issue.

**Strengths:**

This paper proposes a surprising and potential problem in model-based offline RL;

**Weaknesses:**

1. Though this paper points out a novel problem that may be existed in offline RL, the theoretical and empirical evidence seem to be confusing;
2. This proposed approach, RAVL, does not appear to be reasonable for this particular edge-of-reach problem.

see below for details.

**Questions:**

1. This paper use two similar expressions, 'out-of-sample' and 'out-of-distribution', are these two expressions the same or different?
2. According to this paper, the edge-of-reach problem seems to be due to specific interactive schemes of some particular RL tasks, instead of applying to model-based or model-free settings in general.
3. It's confusing that, Table.1 aims to illustrate that it is the edge-of-reach problem that  causes the failure for model-based offline RL methods, while the simple experiment in Figure.2 don't include any model-based methods (SAC is a typical model-free methods).
4. About the proposed method, it's just the previous EDAC algorihtm which is trained with additional sythnesis rollouts through the learned dynamic model. So I can't understand how it can sovle the edge-of-reach issue. It's said to avoid overestimations at edge-of-ereach states, however, the proposed method imposes pessimisic estimation on all training data (according to Eq.3), without distinguishing whether the samples belong to edge-of-reach states or not.

---

> ### Author Response · Authors · 2023-11-15
> **Response to Reviewer 1Cj8**
>
> We would like to thank the reviewer for their time in reviewing our submission. We are encouraged that you noted we tackled a “surprising problem” in model-based offline RL.
>
> **(Question 1) Clarifying out-of-sample and out-of-distribution**
>
> We refer the reviewer to Footnote 1 on page 3 of the main paper where we clarify that we use “out-of-sample” and “out-of-distribution” to mean the same thing. We chose to use both terms to highlight connections to past work. IQL [1], for example, contains a very clear explanation of the model-free version of these issues in which they use the term “out-of-sample”, while many other papers (and wider literature in general) use the term “out-of-distribution”.
>
> **(Question 2) Generality of our finding**
>
> Thank you for raising this question. Theoretically, the edge-of-reach problem may be an issue whenever the effective training dataset (in this case $\mathcal{D}_{\text{rollouts}}$) consists of truncated trajectories that are unable to cover or “reach” the full state space. This is almost certainly the case with the MBPO procedure used in many prominent offline model-based methods (MOPO [2], MOReL [3], MOBILE [4]), combined with limited size offline datasets. Empirically, Table 1 and Figure 5 show that the edge-of-reach problem does indeed occur on the main offline RL benchmark (D4RL), and our new results on V-D4RL *(see part B of our general response)* indicate wider prevalence of the edge-of-reach problem even for latent-space model-based methods. We give a 1-page summary in Appendix A which concisely explains the conditions which induce the edge-of-reach problem.
>
> **(Question 3) Clarification of SAC in Figure 2**
>
> We believe there may be a misunderstanding of the meaning of “SAC” in Figure 2. This figure is designed to illustrate that the MBPO procedure [4] (which is the base algorithm in most prominent offline model-based methods) causes the edge-of-reach problem. The MPBO procedure can be described as:
>
> 1. generate a dataset of truncated rollouts according to the agent’s policy, and then
>
> 2. train the agent (most papers use SAC) on this dataset of rollouts ($\mathcal{D}_{\text{rollouts}}$), and then repeat.
>
> We therefore simply use “SAC” to refer to the base MBPO+SAC procedure (used in MOPO, MOReL, MOBILE…) without our RAVL modification. Please see Table 8 for a summary of the setups. Your question has also prompted us to add Algorithm 1, so thank you for this valuable feedback.
>
> **(Question 4) How does RAVL distinguish between states that are edge-of-reach or not?**
>
> Thank you for this question. We have included a detailed step-by-step walk-through of the intuition in Part A of our general response. Please let us know whether it is still unclear. We would be happy to provide further explanation.
>
> ---
>
> We once again thank you for your valuable feedback and queries which have helped us to improve the paper. We hope our responses were able to clarify your questions. We would be glad to answer any further questions during the discussion period, so please do not hesitate to let us know if you have remaining concerns or if anything is still unclear. If we have been able to address your concerns, we would humbly ask if you would consider raising your score.
>
> ---
>
> [1] Offline Reinforcement Learning with Implicit Q-Learning. Ilya Kostrikov, Ashvin Nair, Sergey Levine. ICLR 2021.
>
> [2] MOPO: Model-based Offline Policy Optimization. Tianhe Yu, Garrett Thomas, Lantao Yu, Stefano Ermon, James Zou, Sergey Levine, Chelsea Finn, Tengyu Ma. NeurIPS, 2020.
>
> [3] MOReL: Model-Based Offline Reinforcement Learning. Rahul Kidambi, Aravind Rajeswaran, Praneeth Netrapalli, Thorsten Joachims. NeurIPS, 2020.
>
> [4] Model-Bellman Inconsistency for Model-based Offline Reinforcement Learning. Yihao Sun, Jiaji Zhang, Chengxing Jia, Haoxin Lin, Junyin Ye, Yang Yu. ICML, 2023.

---

> ### Author Response · Authors · 2023-11-21
>
> Dear Reviewer 1Cj8,
>
> Thank you again for taking the time to review our paper. As the author-reviewer discussion period is about to end (22nd Nov AOE, approximately two days time), we were wondering whether you could let us know if our clarifications were able to answer your questions and clear up any misunderstandings about our algorithm or methodology.
>
> Please let us know if you have any remaining queries or concerns. We hope that we have addressed your questions, and if so, would appreciate it if you would consider revising your score.
>
> Best wishes,
>
> The authors.

---

### Author Response · Authors · 2023-11-15
**General Response - Part 1**

We would like to thank the reviewers for their time reviewing our submission and providing insightful feedback that has helped us improve our paper. We are pleased that reviewers agreed our work was “insightful” and “is likely to have a substantial impact on future algorithmic work in this area”.

We note shared questions about why we should expect the method used in RAVL to address the edge-of-reach issue (reviewers 1Cj8, RQph, 34p6), and also requests for evaluation on a more challenging benchmark (reviewers y8bf, RQph, 34p6). We will address these first in the general response before moving to individual concerns.

## **A - How does RAVL distinguish and apply pessimism to edge-of-reach states?**

We thank the reviewers for asking this question. The intuition is as follows:

- *Reminder: Edge-of-reach states are those that can only ever be sampled as the final next state $s’$ in rollouts.* This means they can appear as next states $s’$ in **$\mathbf{\mathcal{D}_{\text{rollouts}}}$**, but **never as states $\mathbf{s}$ in $\mathbf{\mathcal{D}_{\text{rollouts}}}$**. They may therefore be viewed as **out-of-distribution with respect to the set of states $\mathbf{s}$ in $\mathbf{\mathcal{D}_{\text{rollouts}}}$**.
- The **set of states $\mathbf{s}$ in $\mathbf{\mathcal{D}_{\text{rollouts}}}$** forms the dataset of state inputs that the Q-function is trained on. Therefore edge-of-reach states are those that are **out-of-distribution with respect to the training distribution of the Q-function**.
- As is common in uncertainty estimation literature, high variance over a deep ensemble can be used to detect out-of-distribution inputs. In this way, RAVL’s Q-ensemble is able to detect edge-of-reach states as those that have higher variance over the Q-ensemble due to being out-of-distribution with respect to the training data. Taking the minimum over the ensemble effectively applies a penalty based on ensemble variance (a measure of “out-of-distribution-ness”).
- We verify that RAVL is able to capture this in Figure 3, in which we show that the effective penalty applied by RAVL (the variance over the Q-ensemble) is significantly higher for edge-of-reach states compared to states that are within-reach. (We have edited the paper and caption of Figure 3 to try to make this clearer).

We would like to encourage readers to refer to Appendix A, for more detail on this problem and some more connections which may be helpful. We have also added Algorithm 1 and Table 4 in the Appendix to try to improve clarity.

Please let us know if anything is still unclear. We would be more than happy to engage in further discussion as this is a valuable question and forms the crux of understanding our method.

**(General response continued in next comment.)**

---

> ### Author Response · Authors · 2023-11-15
> **General Response - Part 2**
>
> **(Continued from above.)**
>
> ## **B - Additional experimental evaluation**
>
> **Experiments on Offline Pixel-Based Benchmark (V-D4RL)**
>
> We have run additional initial experiments on the challenging offline *pixel-based* V-D4RL benchmark [1], and are pleased to be able to report that adding RAVL’s edge-of-reach detection and penalization method results in a significant improvement over the baseline on several environments (see Table below).
>
> |  **Environment** |        | **Offline DreamerV2** | **Offline DreamerV2 with RAVL (N=5)** |
> |:----------------:|:------:|:---------------------:|:-------------------------------------:|
> | walker-walk      | medium | 34.1±19.7             | 48.6±8.7                              |
> |                  | medexp | 43.9±34.4             | 47.6±18.3                             |
> | cheetah-run      | medium | 17.2±3.5              | 18.9±3.5                              |
> |                  | medexp | 10.4±3.5              | 13.7±4.5                              |
> | **Overall gain** |        |                       | **+22%**
>
> We observe that RAVL gives a strong boost in performance (+22% overall) on the medium and medexp level datasets, while not helping in the more diverse random and mixed datasets. This observation fits with our intuition of the edge-of-reach problem: the medium and medexp level datasets are likely to have less coverage of the state space and thus we would expect them to suffer more from edge-of-reach issues and the “edge-of-reach seeking behavior” we demonstrated in Section 5.2 of the paper. We note also that these results are without the ensemble diversity regularizer from EDAC [2] (which RAVL uses on the D4RL benchmark), which we anticipate may further increase performance.
>
> Our result on the V-D4RL is particularly notable as the base algorithm used in this pixel-based setting is DreamerV2 [3] (which generates model rollouts in an imagined latent space) rather than MBPO (which uses rollouts in the original state-action space). This suggests that the edge-of-reach problem is more widespread than we had previously indicated.
>
> ----
>
> [1] Challenges and Opportunities in Offline Reinforcement Learning from Visual Observations. Cong Lu, Philip J. Ball, Tim G. J. Rudner, Jack Parker-Holder, Michael A. Osborne, Yee Whye Teh. TMLR, 2023.
>
> [2] Uncertainty-Based Offline Reinforcement Learning with Diversified Q-Ensemble. Gaon An, Seungyong Moon, Jang-Hyun Kim, Hyun Oh Song. NeurIPS, 2021.
>
> [3] Mastering Atari with Discrete World Models. Danijar Hafner, Timothy Lillicrap, Mohammad Norouzi, Jimmy Ba. ICLR, 2021.

---

### Meta-Review · Area_Chair_XjL3 · 2023-12-11

**Metareview:**

This paper studies offline model-based reinforcement learning and uncovers two major issues: the failure of existing algorithms when true dynamics are provided, and the overestimation of values for states only reachable in the final step of limited horizon rollouts. These insights lead to the proposal of Reach-Aware Value Learning (RAVL), a value-based algorithm that addresses value uncertainty at edge-of-reach states. There are some weaknesses of the paper raised from the review comments and discussions, including the weak support of theoretic and empirical evidence, the narrow envs of the experiments, the gap between the theory and the algorithm, and the presentation. Although the authors provided detailed feedbacks, some of the concerns raised are still unsolved.

**Justification For Why Not Higher Score:**

There are some weaknesses of the paper raised from the review comments and discussions, including the weak support of theoretic and empirical evidence, the narrow envs of the experiments, the gap between the theory and the algorithm, and the presentation. Although the authors provided detailed feedbacks, some of the concerns raised are still unsolved.

**Justification For Why Not Lower Score:**

N/A

---

### Decision · Program_Chairs · 2024-01-16

Reject